# Hot Compression Deformation Behavior and Microstructure of As-Cast and Homogenized AA2195 Al-Li Alloy

**Jin Zhang** [1,2], **Zemeng Liu** [1,2] **and Dongfeng Shi** [1,2,*]

1   Light Alloy Research Institute, Central South University, Changsha 410083, China
2   State Key Laboratory of High Performance and Complex Manufacturing, Central South University, Changsha 410083, China
*   Correspondence: 18883726250@163.com

**Abstract:** To understand the effect of the initial state of AA2195 Al-Li alloy on the forming process, as-cast and homogenized ingots were compressed by using a Gleeble-3150 thermo-mechanical simulator at different temperatures (360–480 °C) and strain rates (0.01–10 s$^{-1}$). The hot compression deformation behaviors and microstructural characteristics of the two types of ingots were systematically investigated. The as-cast alloy possessed a better hot compressibility with higher power dissipation efficiency and lower rheological stress than the homogenized alloy under the same deformation conditions. When the temperature was increased above 450 °C, all the alloys showed similar rheological curves. Based on the rheological stress curves, processing maps for the as-cast (AC) and homogenized (HG) alloys were established, and optimal processing domains were identified. In addition, the homogenized alloys were dominated by a fibrous microstructure during deformation, whereas the as-cast alloy produced fine crystals at low temperature (360 °C) and equiaxed crystals at high temperature (480 °C). Our results show that it is possible to use the as-cast 2195 Al-Li alloy as the initial billet to get complicated components. This is attributed to the dispersed eutectic phases, which can provide more nucleation sites for Dynamic Recrystallization (DRX) and Dynamic Recovery (DRV) during hot deformation.

**Keywords:** 2195 Al-Li alloy; hot compression; processing map



## 1. Introduction

With rapid development in the aerospace field, there is increasing demand for high performance and light weight aerospace components. Replacing traditional aluminum alloy with third-generation Al-Li alloys for the manufacture of launch vehicle tanks and aircraft skins can achieve a weight reduction of 10–20% and a strength increase of 15–20% [1–4]. Especially, 2195 Al-Li alloy, which is the representative of the third generation Al-Li alloy, has the advantages of low density, high strength, and high fracture toughness. These excellent properties are accelerating the drive for 2195 Al-Li alloys to replace traditional aluminum alloys in the new aerospace field [5–7]. Recently, a large number of researchers have investigated the microstructural evolution and hot processing properties of 2195 Al-Li alloy [8,9], and the results have shown that the parameters, such as the temperature, and strain rate, as well as precipitates significantly affect recovery and recrystallization during the hot deformation process. Besides this, work hardening and dynamic softening occur simultaneously, which leads to complex deformation behavior of the materials [10,11]. In addition, balance between the work hardening and dynamic softening is key to obtaining good forming capacity [12–14]. In this regard, numerous studies have been done on softening mechanisms during hot compression deformation, among which Nayan et al. [15] explored the microstructural evolution of the forged AA2195 alloy during planar thermal compression. The results illustrated that the main softening mechanism at low temperature was DRV, while at higher temperature, DRX played an extremely important role in the softening behavior. Meanwhile, the authors also concluded that grain size can be controlled by

optimizing parameters to get a better hot workability in AA2195. Yan et al. [16] discovered that fine equiaxed grains are generated by discontinuous dynamic recrystallization during the hot extrusion process in 2195 alloy, excluding continuous dynamic recrystallization. Lu et al. [17] investigated the effect of the $T_1$ phase on the hot compression process of 2195 Al-Li and the results showed that coarse second-phase particles distributed along the grain boundaries facilitate DRX behavior and, finally, affect grain refinement. In addition, Li et al. [18] also observed that the distribution of particles in Al-Cu-Li alloys influence recrystallization during deformation. Zhang et al. [19] investigated the DRX mechanisms of 2195 Al-Li in medium and high temperature compression, and systematically studied the following three types of mechanisms: discontinuous dynamic recrystallization, continuous dynamic recrystallization and geometric dynamic recrystallization. The plasticity and high resilience of 2195 Al-Li alloys at room temperature leads to cracking in the early deformation stage and, then, results in poor hot formability [20]. In order to improve the forming capability of the materials, a large number of investigations have been conducted. Xu et al. [21] investigated the hot deformation behavior of 2195 Al-Li alloy in the secondary hot extruded state and concluded that the deformation activation energy of extruded 2195 Al-Li alloy is much lower than that of spray deposited 2195 alloy. They attributed this to the presence of high mobility LAGBs. Wang et al. [22] found that DRV is accelerated in the regions of low power dissipation at low strain rates in the thermal processing map of 2195 Al-Li alloys.

Most of the previous studies on the hot deformation behavior of 2195 Al-Li alloy have just focused on how to optimize the forming process, but have neglected the material itself, especially the study of the hot deformation behavior of 2195 Al-Li alloy ingots. In order to investigate the effect of the initial material conditions on formability, hot compression experiments were conducted on as-cast and homogenized 2195 Al-Li alloys. The corresponding processing maps were established based on the experimental results, and the rheological behaviors and microstructural evolutions between the two types of alloy ingots were systematically analyzed. The results of the study can provide a reference for cogging and one-step forming of 2195 Al-Li alloy.

## 2. Materials and Methods

### 2.1. Materials

The raw material of AA2195 Al-Li alloy ingots, with a diameter of 190 mm, was fabricated by melting and casting (supplied by Light Alloy Research Institute, CSU), the chemical composition of which (Measured by ICP spectrometer, SPECTRO BLUE SOP, Kleve, Germany) is listed in Table 1. The samples for experiments and observations were cut from the cross section of the ingots at a position three quarters of the way from the center of the initial ingot. Then, a part of this batch of samples was left untreated and the other part was homogenized. The ingot was homogenized at 500 °C for 26 h and then air-cooled to get homogenized HG 2195 samples. Two state samples were taken separately for microstructure observation using OM (model XJP-6A), and the initial microstructure of AC and HG Al-Li alloys are shown in Figure 1a,b, respectively.

According to the statistics of OM results, it can be seen that the two samples had similar average grain size (93–96 μm for 2195 AC Al-Li alloy and 100–105 μm for AA2195 HG Al-Li alloy), and a large number of dendrites were clearly observed in 2195 AC Al-Li alloy, which formed during the solidification process. By contrast, the dendrites disappeared in AA2195 HG Al-Li alloy, and the grains existed as equiaxed crystals. Therefore, after the homogenization treatment, the compounds decreased significantly and the segregated phases dissolved into a matrix.

**Table 1.** Chemical compositions (wt%) of the AA2195 alloy.

| Li | Cu | Mg | Mn | Zn | Zr | Ag | Al |
|------|------|------|------|------|------|------|------|
| 0.90 | 4.00 | 0.28 | 0.04 | 0.03 | 0.13 | 0.28 | Bal. |

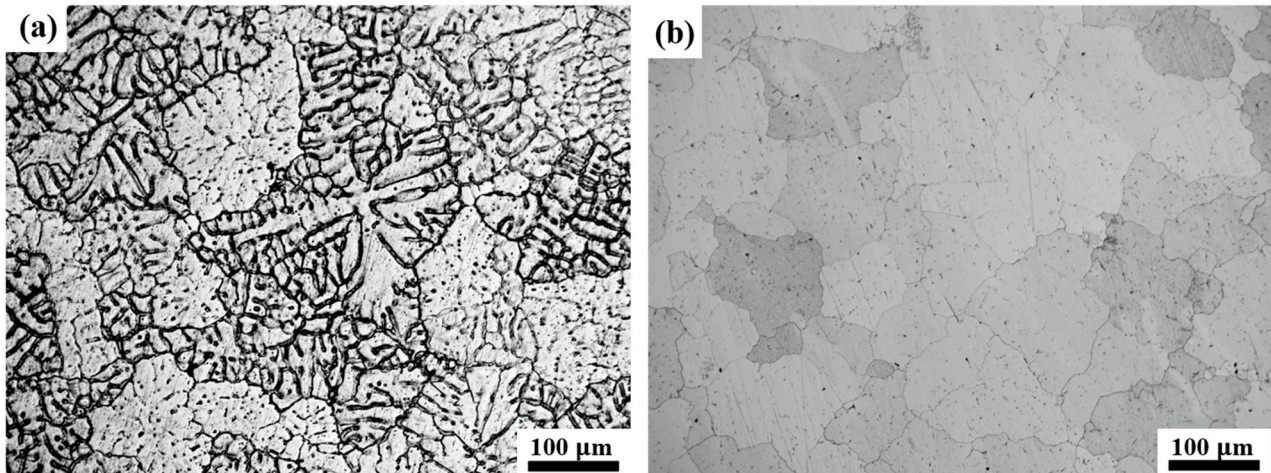

**Figure 1.** Initial microstructures of the AC and HG AA2195 Al-Li alloys: (**a**) AC 2195 Al-Li alloy, (**b**) HG 2195 Al-Li alloy.

*2.2. Experimental Methods*

Hot compression experiments were performed using a Gleeble-3150 (Dynamic System Inc. USA) thermo-mechanical simulator. Based on the ASTM-E9 standard [19], AC and HG Al-Li alloy ingots were processed into cylindrical samples ($\phi$8 mm $\times$ 12 mm). Hot compression tests were conducted on all samples with total deformation of 75%, between 360 °C and 480 °C with increments of 30 °C, and strain rates in the range 0.01 to 10 s$^{-1}$. A graphite lubricant was applied on both ends of the cylindrical samples to reduce friction during compression. The specimens were heated at a rate of 5 °C/s until reaching the target temperature and were then held at this temperature for 3 min before hot compression. Then, the compression tests were stopped at true strain of 1.3% and the samples were directly water-quenched. The schematic diagram of the hot compression process is shown in Figure 2. The true stress–true strain curves were acquired from compression tests and the processing maps for the AC and HG AA2195 Al-Li alloys were established.

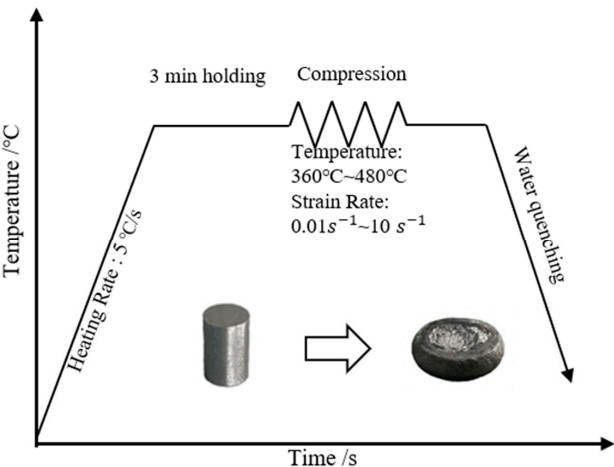

**Figure 2.** Hot compression process.

To study the microstructure evolution of the AC and HG AA2195 Al-Li alloys before and after deformation, the samples were subjected to SEM, EBSD and TEM observations after the tests. The samples for SEM observation were first mechanically ground and then polished with abrasive paper. The EBSD samples were ground, polished and then electropolished using an electrolytic solution, which was a mixture of $HClO_4$ and $C_2H_6O$ (1:9 in volume), using a voltage of 20 v. SEM and EBSD measurements were conducted by using a scanning electron microscope (Helios NanoLab 600i, FEI, Lincoln, NE, USA)

equipped with an HKL EBSD system, Aztec and the Channel 5.0 analysis software. A scanning step size of 1.2 μm at a voltage and a beam current of 2.7 nA, as well as the magnification of 200, was used to acquire the microstructural characteristics. The samples for TEM observation projection were firstly thinned to below 80 μm, and then thinned using a point solution double spraying instrument with a mixture of $HNO_3$ and $CH_4O$ (3:7 in volume) at a voltage of 15 v. The temperature during the twin-jet electro-polished process was controlled to be in the range of $-30\ ^{\circ}C$ to $-25\ ^{\circ}C$. Then, the microstructure of the compressed material was observed using a Tecnai-G220 (FEI, Lincoln, NE, USA) projection electron microscope with an accelerating voltage of 300 kv. In this study, the grain boundaries with a misorientation larger than 15° and misorientation between 2° and 15° were defined as high-angle grain boundaries (HAGBs) and low-angle grain boundaries (LAGBs), respectively.

## 3. Results

### 3.1. Hot Compression Deformation Behavior

According to reference material in [23], it was evident that the friction and lubrication during hot compression had less influence on the error and overall trend of the stress–strain curve. Hence, the original data were used to analyze the hot deformation behavior of 2195 Al-Li alloy. The true stress–true strain curves of AC and HG AA2195 Al-Li alloys under different temperatures and strain rates are shown in Figures 3a–d and 4a–d, respectively. It can be seen that the curves increased to high values of accumulating dislocation at the early deformation stage. With further strain, the curves became invariable and achieved a stable state, owing to dynamic balance of the hardening and softening mechanisms. When the strain rate increased or the temperature decreased, the curves showed higher peak stress and final stable stress, because the frequency of cross-slip and grain migrations reduced at low temperature and strain rate. As deformation reached a stable stage, the AC AA2195 Al-Li alloy had lower rheological stresses compared with the HG alloy, which meant it had better hot compression performance.

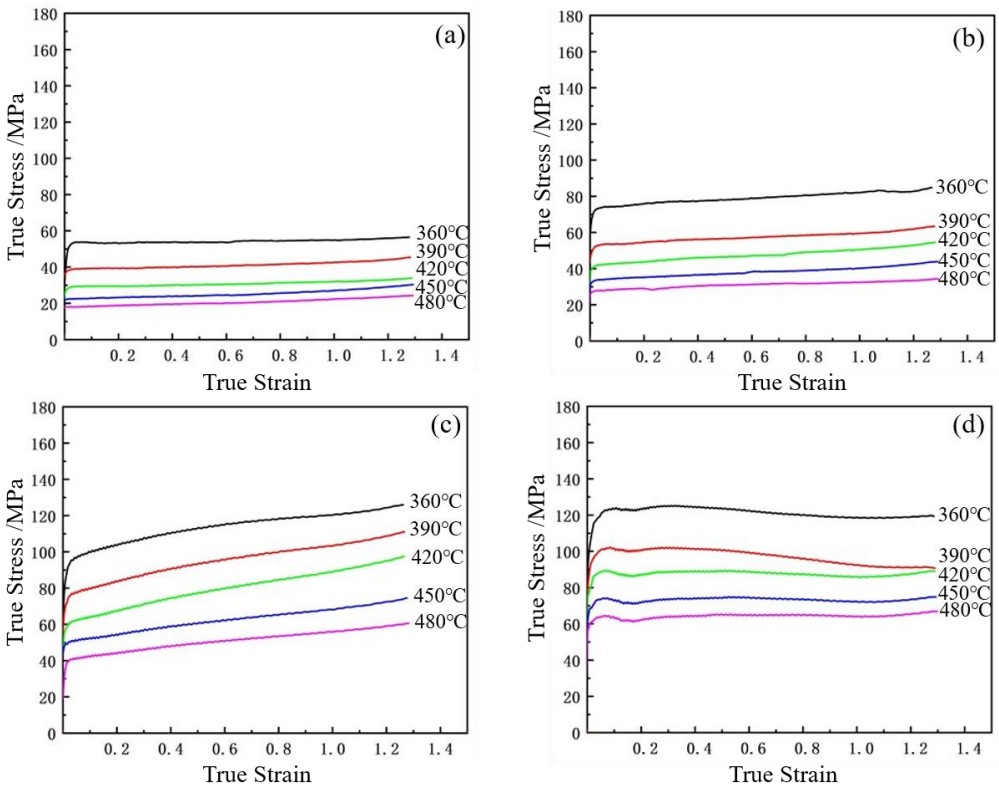

**Figure 3.** True stress–true strain curves for hot compression deformation of AC AA2195 Al-Li alloy at different temperatures and strain rates: (**a**) 0.01 s$^{-1}$, (**b**) 0. 1 s$^{-1}$, (**c**) 1 s$^{-1}$, (**d**) 10 s$^{-1}$.

The curves of AC AA2195 alloy rose to a high value during the initial deformation period, and then the curves gradually increased to a constant value. In HG AA2195 alloy, the curves reached the peak value first then decreased to a steady value. The different trends for the two alloys were due to the fact that the interaction between alloy elements and dislocation was of higher contribution in the HG AA2195 alloy than in the AC AA2195 alloy at the early deformation stage. Furthermore, the work hardening mechanism dominated the deformation process in the AC AA2195 alloy, while the dynamic softening mechanism governed the compression behavior in the HG AA2195 alloy. Therefore, the rheological stresses of the two kinds of alloy show contrary trends.

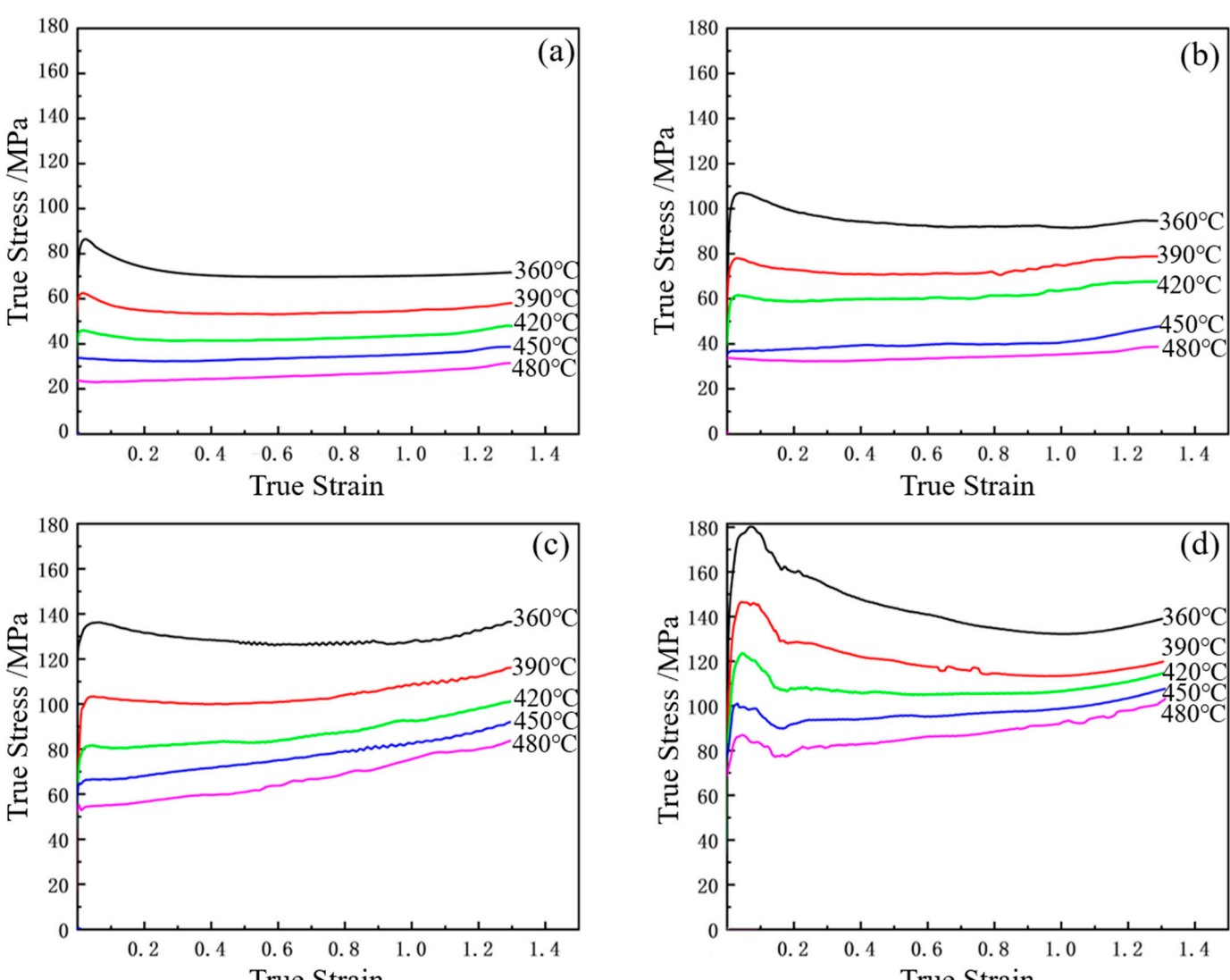

**Figure 4.** True stress–true strain curves for hot compression deformation of HG AA2195 Al-Li alloy at different temperatures and strain rates: (**a**) 0.01 s$^{-1}$, (**b**) 0. 1 s$^{-1}$, (**c**) 1 s$^{-1}$, (**d**) 10 s$^{-1}$.

### 3.2. Differences in Hot Compression Deformation Behavior

The activation energy (Q) represents the size of the energy barrier that the atomic transition needs to overcome, and it is widely used to illustrate the difficulty of material deformation. The activation energy of the material is calculated by the Arrhenius [24–26] equation. The equation is as follows:

$$\dot{\varepsilon} = A\text{F}(\sigma)\exp\left(-\frac{Q}{RT}\right) \tag{1}$$

Under different stress conditions, the equation has three forms:

$$\dot{\varepsilon} = \begin{cases} A_1\sigma^{n_1}\exp(-\frac{Q}{RT}) & \alpha\sigma < 0.8 \\ A\exp(\beta\sigma)\exp(-\frac{Q}{RT}) & \alpha\sigma > 0.8 \\ A[\sinh(\alpha\sigma)]^n\exp(-\frac{Q}{RT}) & \text{for all} \end{cases} \tag{2}$$

where $A$, $A_1$, $n$, $n_1$, $\alpha$ and $\beta$ ($\alpha = \beta/n_1$) are constants, $R$ represents the gas constant, $Q$ represents the activation energy of the material, and $T$ is the deformation temperature, while $\dot{\varepsilon}$ is the strain rate during the experiment, and $\sigma$ is the peak stress. Taking the logarithm of both sides of Equation (2) and then taking the partial derivative of it, the expression for the activation energy $Q$ is as follows:

$$Q = R\left\{\frac{\partial ln\varepsilon}{\partial ln[(\sinh(\alpha\sigma)]}\right\}_T \left\{\frac{\partial ln[(\sinh(\alpha\sigma)]}{\partial\frac{1}{T}}\right\}_t \tag{3}$$

According to the true stress–true strain curves and Equation (3), in order to obtain the effective activation energy values, different strain values were chosen for the calculation. The deformation activation energies of AC and HG AA2195 Al-Li alloys were obtained as $Q_{AC}$ = 170.82–177.44 kJ/mol and $Q_{HG}$ = 176.88–196.12 kJ/mol, respectively. The Q value of HG AA2195 Al-Li alloy was 7–17 kJ/mol higher than that of AC AA2195 Al-Li alloy. The AC AA2195 Al-Li alloy requires less energy for deformation, and is considered to have better hot workability. In order to get a better understanding of the rheological behavior of the materials and, then, to determine the processing domain of the materials, the processing maps of the two alloys were developed, based on the dynamic material model (DMM). The total energy ($P$) is consumed during processing in the form of both plastic deformation of the material (dissipation amount $G$) and microstructural transformations, such as dynamic recovery (DRV) and dynamic recrystallization (DRX) (dissipation co-efficient $J$), where the total energy ($P$) is expressed as:

$$P = G + J = \int_0^{\dot{\varepsilon}} \sigma\,d\dot{\varepsilon} + \int_0^{\sigma} \dot{\varepsilon}\,d\sigma \tag{4}$$

When $G$ and $J$ are obtained in the equation above, the strain rate sensitivity index m can be further obtained, which is calculated as follows:

$$m = \frac{dJ}{dG} = \frac{\partial ln\sigma}{\partial ln\dot{\varepsilon}} \tag{5}$$

The power dissipation coefficient η of the material can be obtained by taking the value of m into Formula (6). The value η is the ratio of the consumed energy to the theoretical maximum consumed energy for characterizing the material microstructure transition during thermal deformation. The higher the value of η is, the more energy is consumed during the microstructural transition, and the material tends to undergo DRX [26].

$$\eta = \frac{J}{J_{max}} = \frac{2m}{m+1} \tag{6}$$

Fitting different η values, strain rate and deformation temperature, the power dissipation efficiency map of AC and HG AA2195 Al-Li alloys were obtained, and the results are shown in Figure 5. In the maps, when the color changes from blue to red, the value η was increasing. Normally, the η values associated with DRX and DRV are considered to be 0.35–0.45 and 0.2–0.3, respectively [24,25]. The results revealed that there was a greater tendency for DRX to occur in the AC AA2195 Al-Li alloy at a strain rate of 0.01–0.1 s$^{-1}$ and temperature of 400–440 °C (η > 0.35), while the HG AA2195 Al-Li alloy was dominated by DRV at temperatures above 440 °C (η < 0.3).

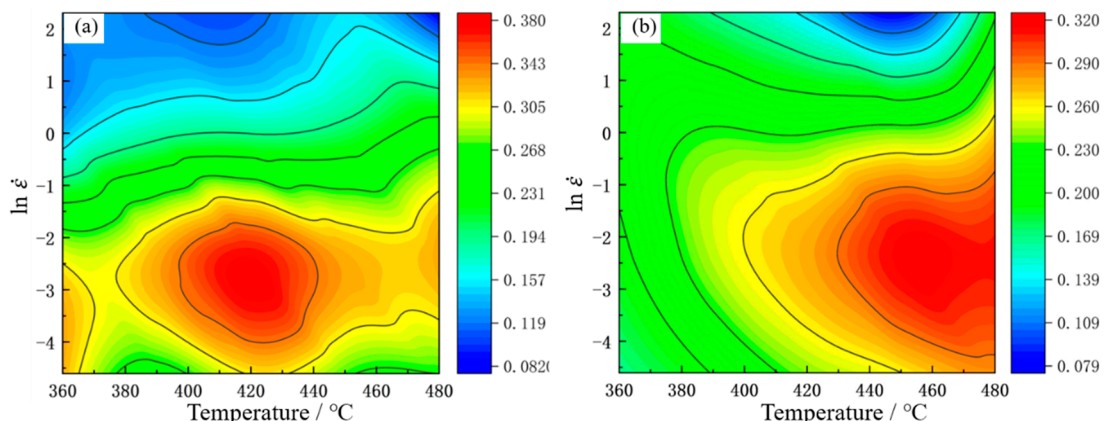

**Figure 5.** Power dissipation efficiency maps of AC and HG AA2195 Al-Li alloys under different thermal deformation conditions: (**a**) AC 2195 Al-Li alloy, (**b**) HG 2195 Al-Li alloy.

According to the maximum entropy generation theory, a destabilization criterion for identifying unstable deformation regions, such as local flow and cracking is established with the following equation:

$$\xi(\dot{\varepsilon}) = \frac{\partial ln\left(\frac{m}{m+1}\right)}{\partial ln\dot{\varepsilon}} + m < 0 \tag{7}$$

The power dissipation coefficient η and the instability criterion $\xi(\dot{\varepsilon})$ were fitted by Origin 9.0 software (The software publisher is OriginLab Corporation, provided by CSU) to draw the processing map of AC and HG AA2195 Al-Li alloy, as shown in Figure 6. The dark area in the processing map indicates the unstable processing region, and the white area indicates the stable processing region. According to the processing maps, the optimal machining regions of AC and HG AA2195 Al-Li alloy were 400–440 °C with a strain rate of 0.01–0.1 s$^{-1}$ and 440–480 °C with a strain rate of 0.01–0.1 s$^{-1}$, respectively. Comparing the respective optimal processing maps of the alloys, the AC alloy demanded a lower temperature, indicating that it required less energy to deform and had a greater deformation capacity.

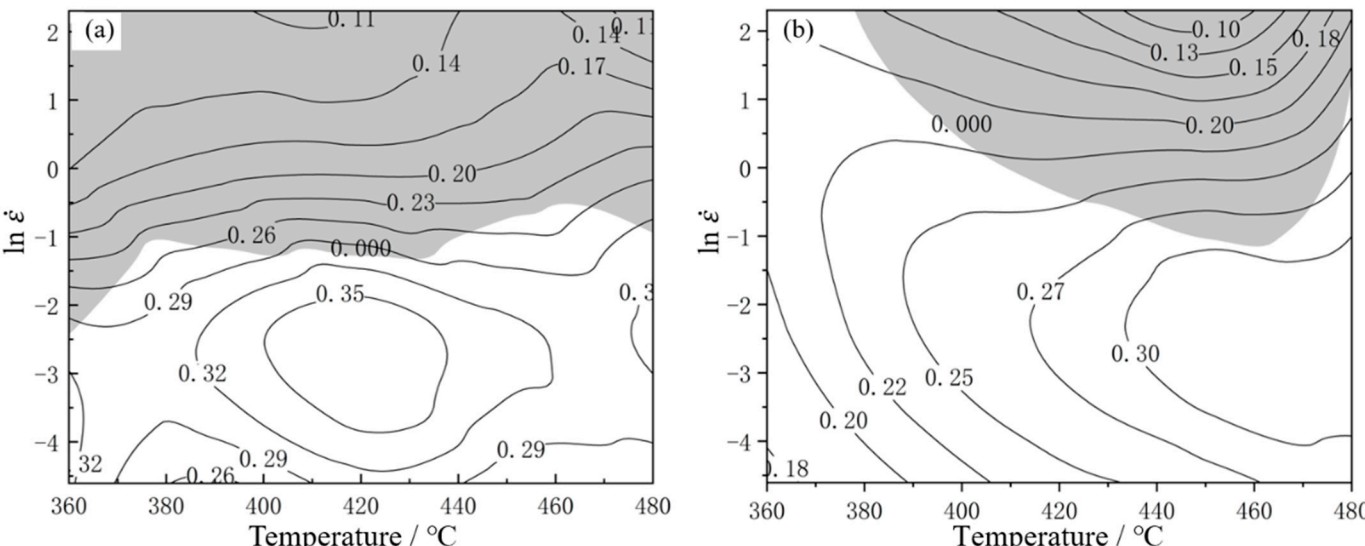

**Figure 6.** Processing map of AC and HG 2195 Al-Li alloys under different hot deformation conditions: (**a**) AC 2195 Al-Li alloy, (**b**) HG 2195 Al-Li alloy.

### 3.3. Microstructure Differences

Figures 7 and 8 exhibit the IPF maps of AC and HG AA2195 Al-Li alloys compressed isothermally with a strain rates range of 0.01 to 10 s$^{-1}$ at temperature of 360 °C, 420 °C and 480 °C. The HAGBs and LAGBs are indicated by black and white lines, respectively. Obviously, a mass of sub-grains formed within grain interiors in all alloys. The number of sub-grains decreased with increase in the deformed temperature from 360 °C to 480 °C or decrease in the strain rate from 0.01 s$^{-1}$ to 10 s$^{-1}$. However, at low temperature, the number of sub-grains in the AC AA2195 alloy was significantly higher than that in the HG AA2195 alloy, which indicated a stronger DRV activity in the AC AA2195 alloy. Besides, as the temperature rose to 480 °C, a fully recrystallized grain structure was easily found in the AC alloy (marked by the black dashed line in Figure 7). It showed that DRX occurred during hot compression of the AC AA2195 Al-Li alloy under this condition. On the contrary, this phenomenon was not obvious in the HG alloy. Apparently, compared with HG AA2195 alloy, the AC AA2195 alloys are more prone to recrystallization under the same conditions, leading to better thermal compressibility.

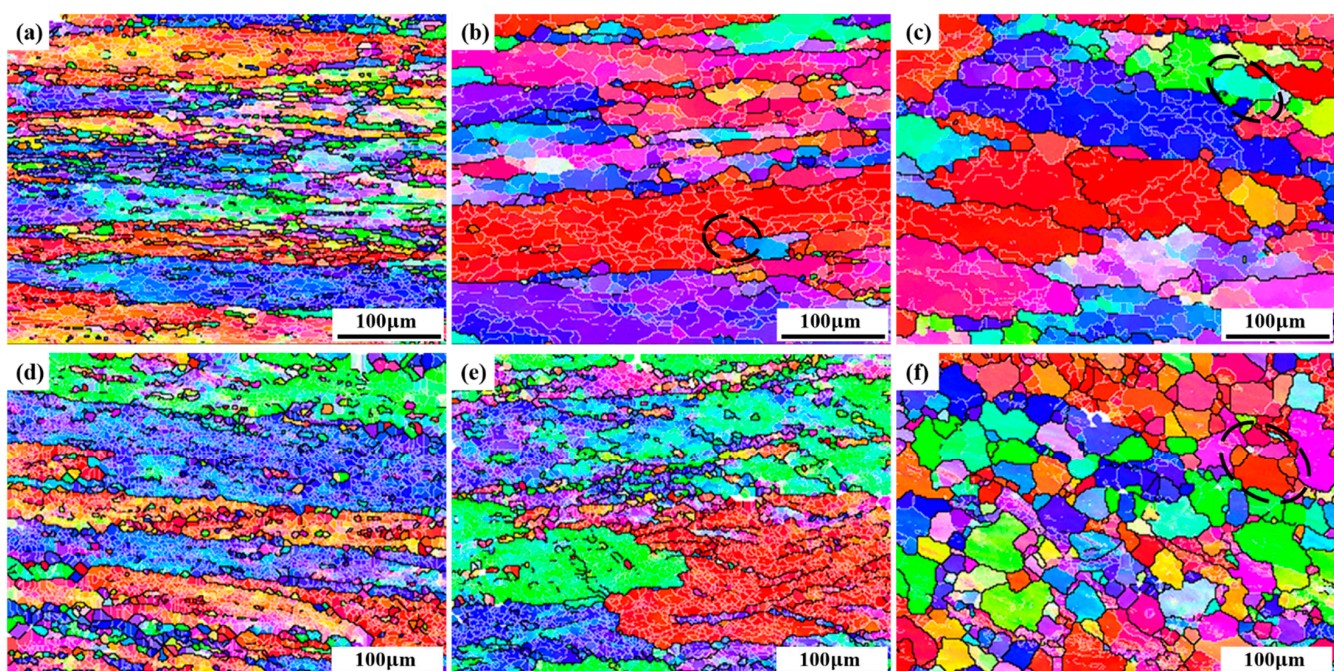

**Figure 7.** IPF maps of the AC AA2195 Al-Li alloy under different thermal deformation conditions: (**a**) 360 °C, 0.01 s$^{-1}$; (**b**) 420 °C, 0.01 s$^{-1}$; (**c**) 480 °C, 0.01 s$^{-1}$; (**d**) 360 °C, 0.01 s$^{-1}$; (**e**) 420 °C, 0.01 s$^{-1}$; (**f**) 480 °C, 0.01 s$^{-1}$.

The SEM micrographs of the AC and HG AA2195 alloys with low and high magnifications are shown in Figure 9a–d. A large number of discontinuous fine residual eutectic phases can be observed in the AC fabric, while, on the contrary, non-equilibrium phases are found in the HG fabric. The eutectic phases dissolved into the matrix after homogenized treatment. In addition, some representative eutectic phases (marked by arrows and numbers) were selected for EDS analysis, and the corresponding results of their compositions are shown in Table 2. According to the EDS results, the eutectic phases were mainly composed of Mg and Cu elements. The effects of these diffused eutectic phases on the DRX and DVR of AA2195 Al-Li alloy is discussed below.

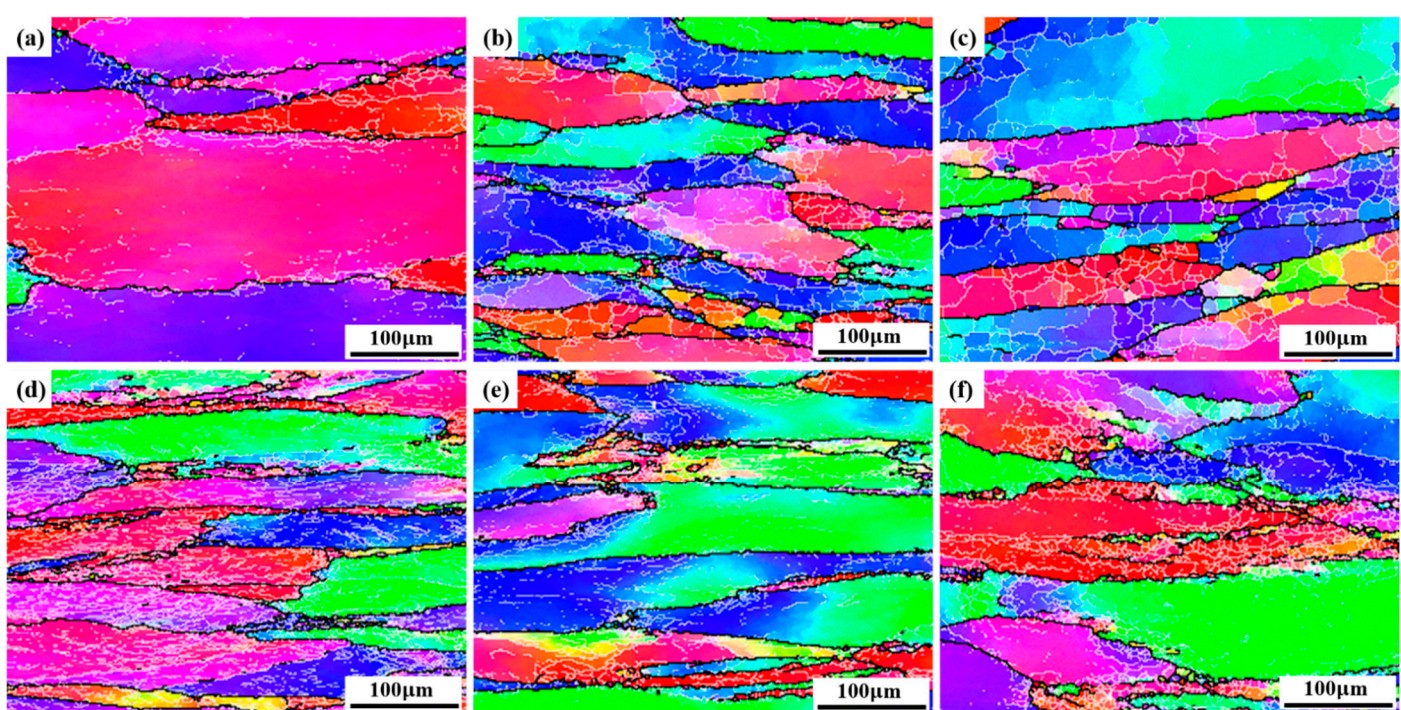

**Figure 8.** IPF maps of HG AA2195 Al-Li alloy under different thermal deformation conditions: (**a**) 360 °C, 0.01 s$^{-1}$; (**b**) 420 °C, 0.01 s$^{-1}$; (**c**) 480 °C, 0.01 s$^{-1}$; (**d**) 360 °C, 10 s$^{-1}$; (**e**) 420 °C, 10 s$^{-1}$; (**f**) 480 °C, 10 s$^{-1}$.

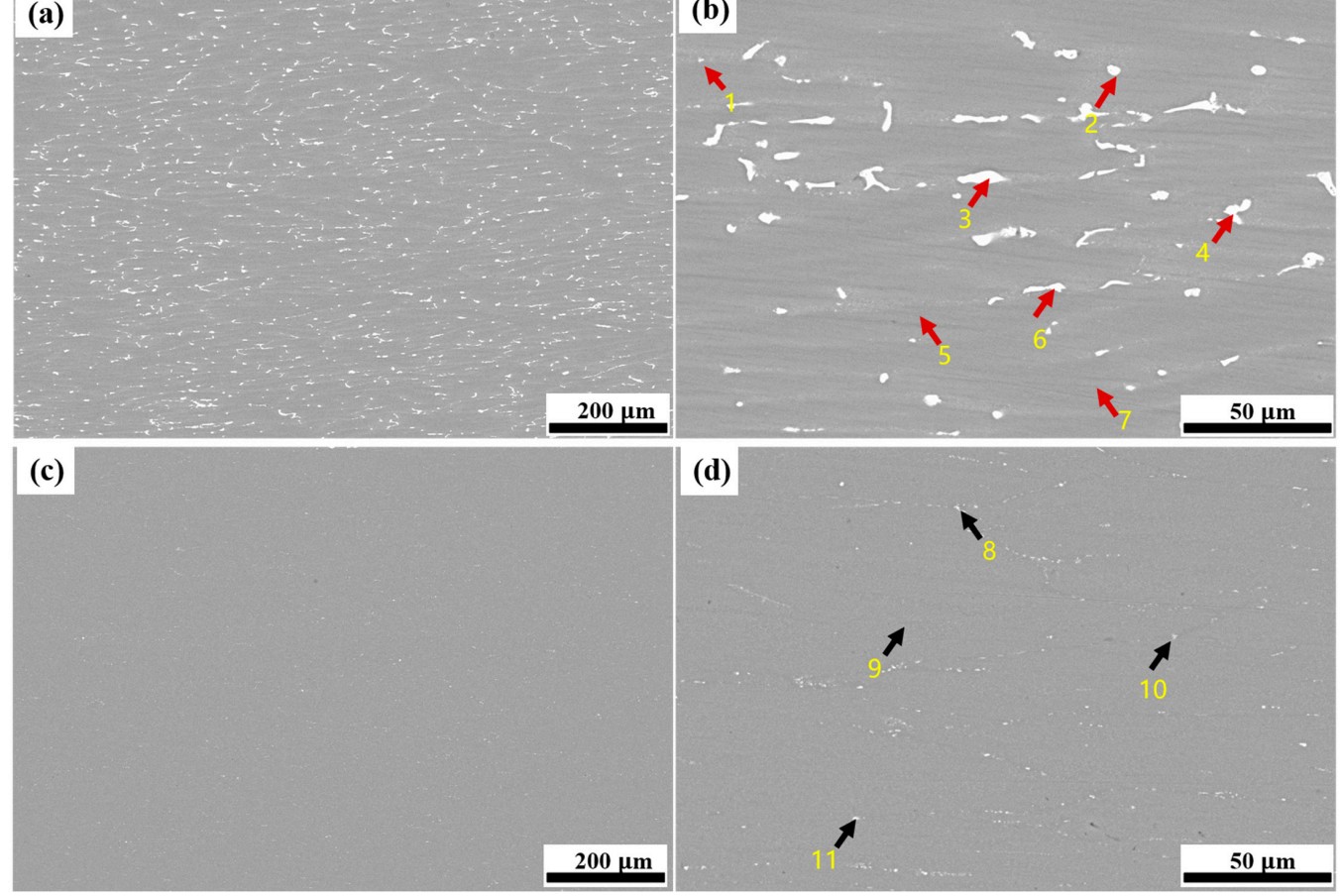

**Figure 9.** SEM images of (**a**,**b**) AC and (**c**,**d**) HG AA2195 Al-Li alloy after hot compression deformation.

**Table 2.** EDS analysis (wt%) on the selected positions in Figure 9b,d.

| Position | Al | Cu | Ag | Mg | Fe | Zn | Zr |
|---|---|---|---|---|---|---|---|
| 1 | 93.1 | 6.2 | 0.1 | 0.5 | - | - | - |
| 2 | 85.3 | 13.7 | 0.1 | 0.7 | 0.1 | 0.1 | - |
| 3 | 85.0 | 13.6 | 0.1 | 0.9 | - | 0.1 | 0.2 |
| 4 | 78.9 | 20.5 | 0.1 | 0.4 | - | - | - |
| 5 | 98.4 | 1.0 | - | 0.4 | - | 0.1 | - |
| 6 | 85.6 | 13.4 | 0.5 | 0.4 | - | - | 0.1 |
| 7 | 98.5 | 1.1 | - | 0.3 | - | - | - |
| 8 | 87.5 | 8.9 | 0.1 | 0.4 | 2.6 | 0.2 | - |
| 9 | 97.9 | 1.6 | 0.1 | 0.4 | - | - | - |
| 10 | 91.6 | 6.9 | 0.1 | 0.5 | - | - | 0.3 |
| 11 | 94.3 | 4.9 | 0.1 | 0.3 | - | - | 0.3 |

## 4. Discussion

To better explain the hot compression properties of AC AA2195 Al-Li alloy, the precipitates, GOS maps and recrystallization of the samples were analyzed for the two alloys after compression. Figure 10 shows the TEM bright field and dark field images of the AC AA 2195 alloy. The areas marked with red dashed lines in the figure are precipitates, and the grains marked with yellow letters are recrystallized grains. It can be seen that there were a large number of precipitates of large size inside the recrystallized grains and out of the grain boundaries. During the deformation process, the matrix bypasses these large precipitates and forms dislocation rings around them. The Al matrix near the precipitates underwent coordinated deformation, which led to an increase in energy storage around the precipitates, and these regions were more prone to recrystallization under large energy storage conditions. The results of TEM showed that some phases of 1–2 um in size were present in the AC AA2195 Al-Li alloy. These Al-Cu phases were distributed in as-cast alloy along the grain boundaries of the recrystallized grains by diffusion. Based on the particle excited nucleation (PSN) theory, in the AC AA2195 alloy, the distributed residual precipitates provide more nucleation sites for DRX and DRV. Therefore, DRX and DRV mechanisms are significantly facilitated and the material softens. Compared to the HG AA2195 Al-Li alloy, this was the main factor for the AC AA2195 alloy to achieve lower rheological stress.

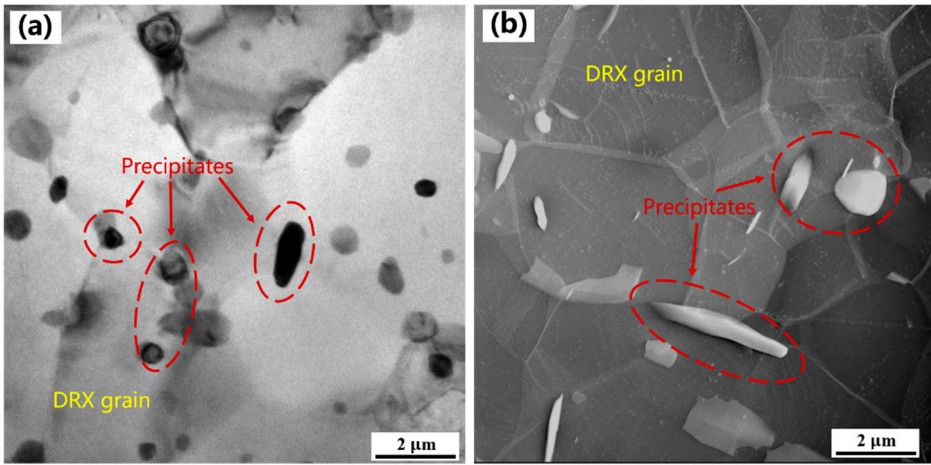

**Figure 10.** TEM image of AC AA2195 alloy: (**a**) is a TEM bright field image, and (**b**) is a STEM-HAADF image.

GOS represents the degree of orientation variation at each point in the crystal structure and the average orientation of the crystal structure can be used to reveal the level of lattice distortion or strain within a grain. Figure 11 shows the grain orientation spread (GOS) plots

of the AC and HG AA2195 alloys after compression in various situations [26]. The color of scale bar gradually changed from blue to red, indicating that the GOS value was gradually increasing. In one grain, the larger the GOS value is, the larger the strain is. Analysis of the GOS results revealed that a great number of fine blue grains close to the grain boundaries were observed in the AC AA2195 alloy at low temperature, and the size and number of blue grains increased with higher temperature. In contrast, in the HG AA2195 alloy, a few blue grains were observed at 360 °C, which indicated that the grains of HG AA2195 Al-Li alloy underwent a large internal deformation. As the temperature increased to 480 °C, only small blue grains were found at the grain boundaries. This suggested that the DRX and DRV of the AC AA2195 alloys were significantly higher than those of the HG AA2195 alloy under the same conditions.

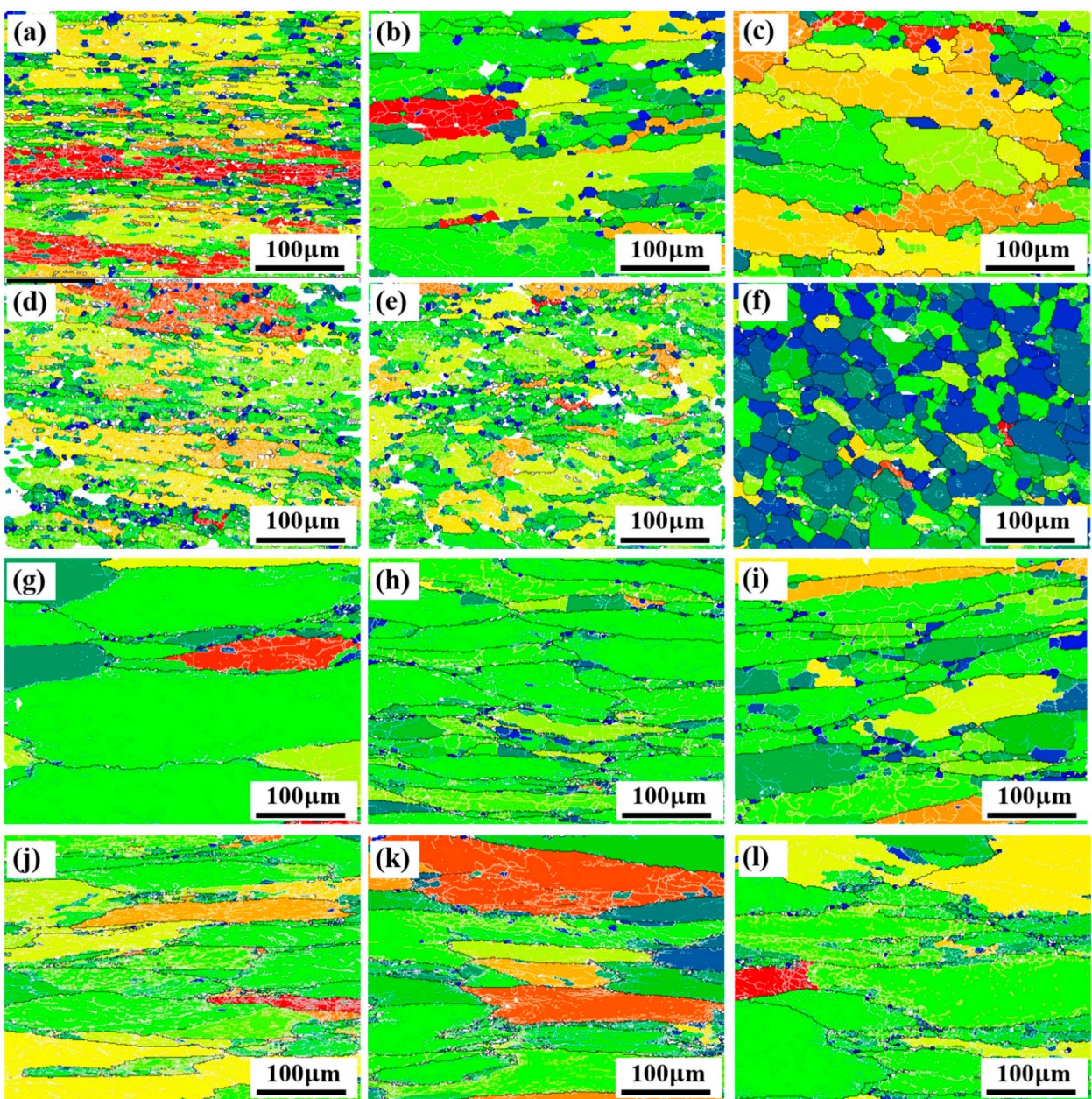

**Figure 11.** Grain orientation spread (GOS) plots of AC and HG AA2195 alloy in different deformation conditions: (**a**) AC-360 °C /0.01s$^{-1}$, (**b**) AC-420 °C /0.01s$^{-1}$, (**c**) AC-480 °C /0.01s$^{-1}$; (**d**) AC-360 °C /10s$^{-1}$, (**e**) AC-420 °C /10s$^{-1}$, (**f**) AC-480 °C /10s$^{-1}$; (**g**) HG-360 °C /0.01s$^{-1}$, (**h**) HG-420 °C /0.01s$^{-1}$, (**i**) HG-480 °C /0.01s$^{-1}$; (**j**) HG-360 °C /10s$^{-1}$, (**k**) HG-420 °C /10s$^{-1}$, (**l**) HG-480 °C /10s$^{-1}$.

Normally, HAGBs are used to represent the recrystallization processing level of samples, and the higher the value is, the more grains of dynamic recrystallization there are [27,28]. The percentages of HAGBs in the alloys in the two states were counted using Channel 5 software, and the results are shown in Figure 12. The results showed that the percentage of HAGBs in the AC AA2195 alloy was much higher than that in the HG AA2195 alloy. These results were evidence that the recovery and recrystallization processing levels of the AC AA2195 alloy were higher than those of the HG 2195 alloy.

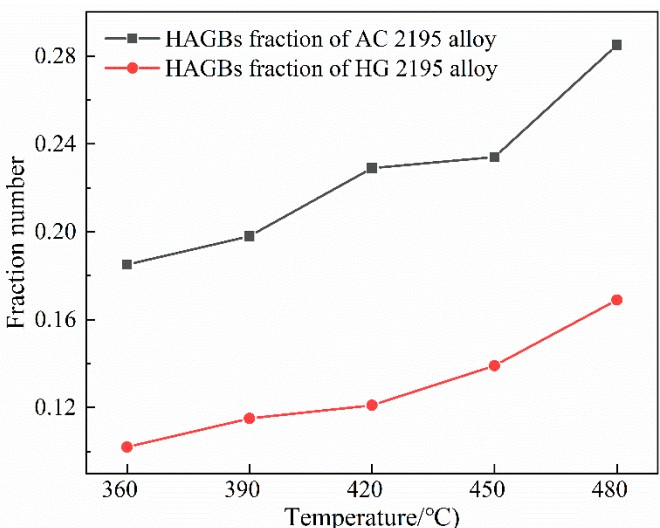

**Figure 12.** The fraction of HAGBs for AC and HG AA2195 alloy under different deformation conditions.

The Kernel Average Misorientation (KAM) reflects the degree of plastic deformation, and a higher value indicates a greater degree of plastic deformation or a higher density of defects. In addition, the distribution of KAM is consistent with the distribution of geometrically necessary dislocations (*GND*) in the material [29]. Therefore, the KAM distribution and recrystallization behaviors were analyzed for the two alloys under different deformation conditions. The grain boundaries misorientation lower than 5° were calculated in all samples. Furthermore, the *GND* density was calculated as shown in Equation (5) [30,31], where $\rho$ is the *GND* density at the test point (in $m^{-2}$), $\theta$ is the local error direction (in radians), *B* is the Bernoulli vector (unit: nm), and u is the step size set for the test (unit: μm).

$$\rho_{GND} = \frac{2\theta}{uB} \tag{8}$$

Figure 13 shows the KAM values for different temperatures and strain rates. The results indicated that the main trend of KAM values for both alloys was a gradual decrease with temperature increase, but the values of the AC AA2195 alloy were still higher than those of the HG AA2195 alloy under the same conditions. This indicated that a more intense deformation occurred internally in the AC AA2195 alloy. Figure 14 exhibits the KAM and corresponding recrystallization images of AC and HG AA2195 Al-Li alloy under different deformation strain rate and temperature. The color in the KAM plot changed from blue to red, which indicated that the local dislocation gradually increased. In the recrystallization plot, the blue regions represent the recrystallized grains during hot deformation. Compared with the HG AA2195 alloy, the recrystallization areas of the AC AA2195 alloy were more obvious. Due to DRX and DRV, the AC AA2195 alloy became more softened and had better hot workability during deformation. From the KAM plot analysis, it can be seen that the KAM decreased with increasing temperature and the dislocation density in the AC AA2195 alloy was higher than that of the HG AA2195 alloy at 420–480 °C, since the DRX of the AC AA2195 alloy increased at higher temperature and the density of dislocations decreased. During the compression process, the grains in the AC AA2195 alloy experienced a process of

dislocation proliferation to recrystallization and then to dislocation proliferation. Therefore, the materials had better hot workability.

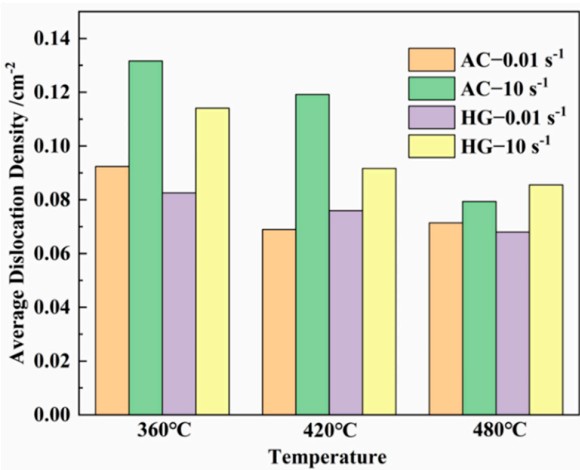

**Figure 13.** KAM values of AC and HG AA2195 alloy under different deformation conditions.

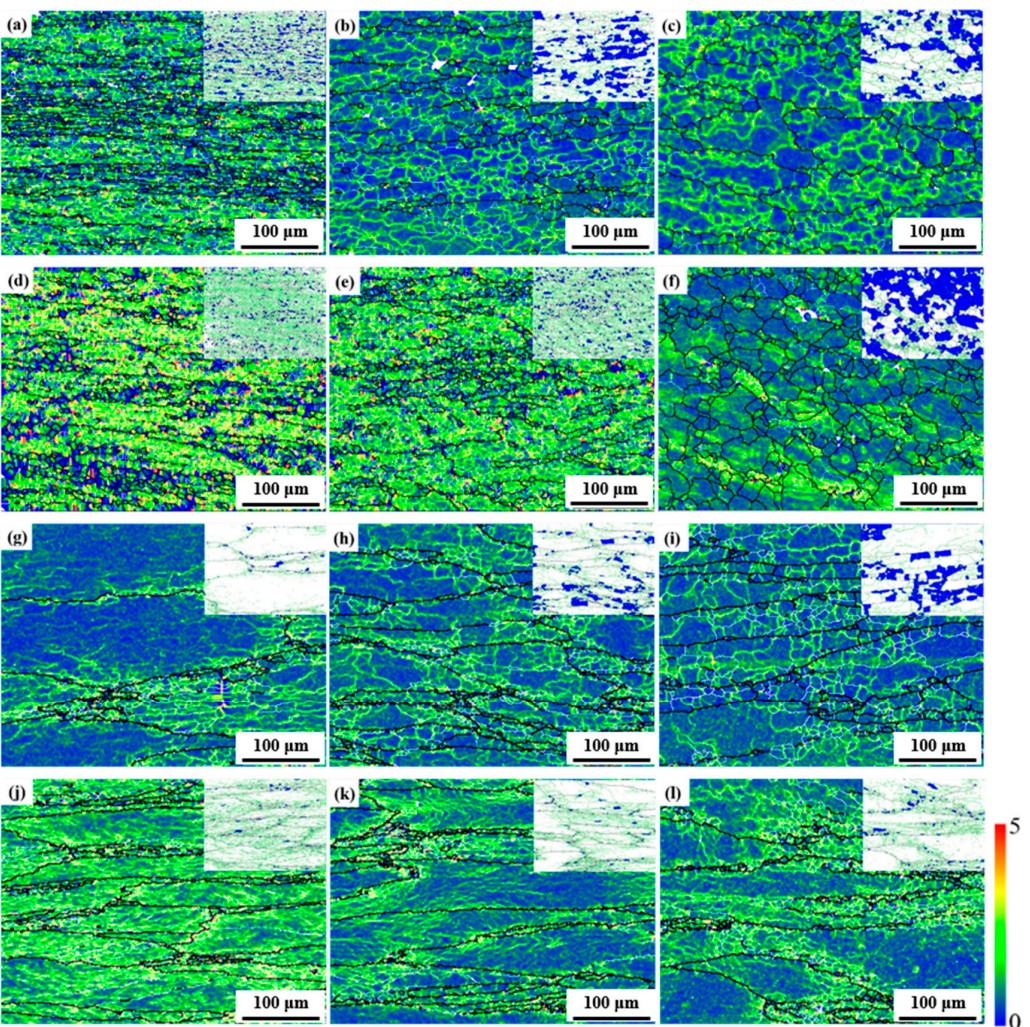

**Figure 14.** KAM and recrystallization of AC and HG AA2195 Al-Li alloy under different hot deformation conditions: (**a**) AC-360 °C /0.01s$^{-1}$, (**b**) AC-420 °C /0.01s$^{-1}$, (**c**) AC-480 °C /0.01s$^{-1}$; (**d**) AC-360 °C /10s$^{-1}$, (**e**) AC-420 °C /10s$^{-1}$, (**f**) AC-480 °C /10s$^{-1}$; (**g**) HG-360 °C /0.01s$^{-1}$, (**h**) HG-420 °C /0.01s$^{-1}$, (**i**) HG-480 °C /0.01s$^{-1}$; (**j**) HG-360 °C /10s$^{-1}$, (**k**) HG-420 °C /10s$^{-1}$, (**l**) HG-480 °C /10s$^{-1}$.

## 5. Conclusions

In this study, compression tests of AC and HG AA2195 Al-Li alloys were conducted at different temperatures (360 °C, 420 °C, 480 °C) and strain rates (0.01 s$^{-1}$, 0.1 s$^{-1}$, 1 s$^{-1}$ and 10 s$^{-1}$). The microstructure evolutions were analyzed and the processing maps were established. The main conclusions of this investigation follow:

During hot compression, the homogenized alloy is dominated by a fibrous microstructure, whereas the as-cast alloy produces fine crystals at low temperature (360 °C) and equiaxed crystals at high temperature (480 °C). The applicable processing condition for the as-cast alloy is the strain range of 0.01–0.1s$^{-1}$ and the temperature range of 400–440 °C, while a strain rate range of 0.01–0.1s$^{-1}$ and temperature of 440–480 °C are the suitable processing conditions for the homogenized state alloy.

The as-cast alloy possessed a better hot compressibility with higher power dissipation efficiency and lower rheological stress than the homogenized alloy under the same deformation conditions. In the AC AA2195 alloy, diffusely distributed residual precipitates provide more nucleation sites for DRX and DRV, and, therefore, the recovery and recrystallization processing levels of the AC AA2195 alloy are higher than those in the HG AA2195 alloy. During hot compression, the AC AA2195 alloy can achieve a better balance of working hardening and dynamic softening, leading to good forming capacity. The as-cast AA2195 Al-Li alloy can act as the initial billet to achieve good hot workability.

**Author Contributions:** Conceptualization, Z.L., D.S. and J.Z; methodology, Z.L.; software, Z.L. and D.S.; validation, Z.L., D.S. and J.Z.; formal analysis, Z.L., D.S. and J.Z.; investigation, Z.L.; resources, Z.L. and J.Z.; data curation, Z.L. and D.S.; writing—original draft preparation, Z.L.; writing—review and editing, D.S. and J.Z.; visualization, J.Z; supervision, D.S. and J.Z.; project administration, J.Z.; funding acquisition, J.Z. All authors have read and agreed to the published version of the manuscript.

**Funding:** This work was funded by the National Key R&D Program of China (grant No. 2020YFA0711104), National Natural Science Foundation of China (grant No. U21B6004).

**Institutional Review Board Statement:** Not applicable.

**Informed Consent Statement:** Not applicable.

**Data Availability Statement:** The data presented in this study are available on request from the corresponding author.

**Acknowledgments:** We gratefully thank Central South University as the author's institutions for their strong support in our conducted research.

**Conflicts of Interest:** The authors declare no conflict of interest.

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
