# Peer review of "Hot Compression Deformation Behavior and Microstructure of As-Cast and Homogenized AA2195 Al-Li Alloy"

_metals, doi:10.3390/met12101580_

Round 1

Reviewer 1 Report

The manuscript can not be accepted in its present form due to the following major comments:

1. The manuscript has poor significance content as well as poor presentation. The whole manuscript has to be revised and re-written carefully. The experimental steps section is not clear, and should be in more details

2. The microstructures characterization has to be improved. The authors should present high quality microstructures to be very well clear to the readers.

3. The conclusions have to be more significant and go well with the main objectives and results of the manuscript.  

Reviewer 2 Report

The manuscript is very interesting; however, I have quite few comments for the authors:

1.      Firstly, grammar check should be considered (some are marked in the manuscript).

2.      In the title of the manuscript the standard should be listed (AA or EN AW).

3.      Fig. 10 does not include FFT therefore you cannot clearly state which phase is it. Please add.

4.      Figure 7: the title is not correct.

5.      Figure 9: a, b, c and d are not explained. Position in Table 2 should be marked in Fig. 9.

6.      Where all the values for the maps calculations were taken?

Reviewer 3 Report

The authors studied the hot deformation behaviour and microstructure of an Al-Li alloy in two conditions. The experimental work consisted of hot deformation tests and the evaluation of results.

The work is rather novel and presented appropriately.

There are several topics on which the article should be improved.

1) Title

The authors studied one alloy. Thus the word "alloys" is inappropriate.

2) Introduction

The introduction is very scarce. It should give a broader overview of the topic. In addition, only few references were given.

3) Materials and methods

a) How was the chemical composition determined?

b) Line 72: No standard deviations are given for the grain sizes. And two decimals for the grain size are inappropriate

4) Results

a) Line 204; details of EDS analysis missing

b) Line 207: In the figure, the caption should explain the meaning of the arrows.

c) Figure 10: The phases should be labelled. No diffraction patterns of the phases present were given.

Sections 4 and 5 have the same name: "Discussion". Section 5 is probably "Conclusions".

Reviewer 4 Report

In the paper " Hot Compression Deformation Behavior and Microstructure of As-Cast and Homogenized 2195 Al-Li Alloys", the authors have constructed the constitutive models and processing maps for the prediction of the hot deformation behaviour of the aluminium alloy. The obtained values of the effective activation energy for as-cast and homogenized 2195 Al-Li Alloys are not sufficient. It means that the hot deformation behaviour of the alloy in both states is similar and the alloy may be deformed in as-cast state at low strain rates. The presented results seem to be interesting. However, some parts of the manuscript are needed to be modified accordingly following comments:

1.                 The friction between the sample’s edges and the dies such as adiabatic heating during the deformation may significantly influence the true stress – true strain curves [10.1016/j.jallcom.2018.08.010, 10.1179/026708301101510843]. The shape of the curves at high strain values shows that the authors did not consider this fact. Using of the graphite lubricant is not enough for the friction reducing.

2.                 Line 139: The obtained values of the effective activation energy for as-cast and homogenized 2195 Al-Li Alloys are not sufficient. The authors should add the error of the energies’ values determination. It should be significantly larger than 4 kJ/mol.

3.                 The authors should provide the strain values for the construction of the constitutive models and processing maps. Usuallym the stress at steady state should be used for the construction. However, steady state is not evident for the part of the stress-strain curves.

4.                 Minor corrections:

-                     Line 86: true strain should be 1.3 but not 1.3%.

-                     Error is in Eq. (6).

-                     Caption for Figure 7 is incorrect.

-                     Scale for figure 10 should be added.

-                     Part 5 should be "Conclusions".

-                     The values of the grain size through the text should be given with confidence intervals. The number of digits after a dot should be decreased accordingly to the value of the confidence interval.

Round 2

Reviewer 1 Report

The manuscript can be accepted in its presnet form. The authours did most of the required modifications. 

Author Response

Thank you for your letter and the comments concerning our manuscript entitled “Hot Compression Deformation Behavior and Microstructure of As-Cast and Homogenized 2195 Al-Li Alloy” (metals-1875502). Those comments are valuable and very helpful. We have read through comments carefully and have made corrections. we have changed the english language.

Reviewer 4 Report

The authors have answered previous comments and improved the manuscript. The paper may be accepted for publication.

Author Response

Thank you for your letter and the comments concerning our manuscript entitled “Hot Compression Deformation Behavior and Microstructure of As-Cast and Homogenized 2195 Al-Li Alloy” (metals-1875502). Those comments are valuable and very helpful.